# The Effect of Ethylene on the Color Change and Resistance to *Botrytis cinerea* Infection in ‘Kyoho’ Grape Fruits

**DOI:** 10.3390/foods9070892

**Published:** 2020-07-07

**Authors:** Tianyu Dong, Ting Zheng, Weihong Fu, Lubin Guan, Haifeng Jia, Jinggui Fang

**Affiliations:** Key Laboratory of Genetics and Fruit development, Horticultural College, Nanjing Agricultural University, Nanjing 210095, China; 2017104011@njau.edu.cn (T.D.); 2015204002@njau.edu.cn (T.Z.); 2018804144@njau.edu.cn (W.F.); 2019804138@njau.edu.cn (L.G.); 2018204002@njau.edu.cn (J.F.)

**Keywords:** grape, ethephon, *Botrytis cinerea*, gene expression, VvERF1

## Abstract

The formation of grape quality and the mechanism of resistance against foreign pathogens affect the storage stability of fruits during post-harvest handling. Ethylene plays a crucial role in regulating the ripeness of fruits and can be used as an exogenous regulator to resist exogenous pathogens. In this study, we used different concentrations of ethephon for treatment of grape fruits before veraison, analyzed the anthocyanin content, soluble solids, titratable acid, and determined fruit firmness and cell wall metabolism-related enzymes during fruit development. Results showed that exogenous ethephon promoted the early coloration of grape fruits and increased the coloring-related genes myeloblastosis A1(MYBA1), myeloblastosis A2(MYBA2), chalcone isomerase (CHI), flavanone 3-hydroxylase (F3H), flavonoid 3’-hydroxylase gene (F3’H), flavonoid 3’, 5’hydroxylase (F3’5’H), 3-O-flavonoid glucosyltransferase (UFGT), and glutathione S-transferase (GST), softening related genes Polygalacturonase(PG), pectinate lyases(PL) and Pectin methylesterase( PME, as well as ethylene metabolism pathway-related genes 1-aminocyclopropane-1-carboxylic acid synthase 1(ACS1), 1-aminocyclopropane-1-carboxylic acid oxidase 2 (ACO2), ethylene receptor gene(ETR2), and ethylene-insensitive 3 (EIN3). Ethephon treatment also increased soluble solids and decreased titratable acid in grape fruit. Fruits pretreated with ethephon were inoculated with *Botrytis cinerea*, which led to resistance in grape fruit through activation of the antioxidant system. The expression levels of disease resistance-related genes including *VvPAD4*, *VvPIP1*, *VvNAC26*, *VvDREB*, *VvAPX*, *Vvpgip*, *VvWRKY70*, *VvMYC2*, *VvNPR1* also increased in inoculated fruit with pathogen following ethephon pretreatment. Furthermore, we monitored ethylene response factor 1(ERF1) transcription factor, which could interact with protein EIN3 during ethylene signal transduction and mediate fruit resistance against *B. cinerea* infection. Meanwhile, overexpression of VvERF1 vectorin strawberry fruits reduced the susceptibility to *B. cinerea* infection. We suggest that ethylene can induce resistance in ripened fruits after *B. cinerea* infection and provide adequate postharvest care.

## 1. Introduction

Grape (*Vitis vinifera*) is the world’s most important economic crops and it is the world’s main fruit tree variety [1]. The grape has high nutritional value and the fruit is rich in many vitamins [2]. During the ripening process of the grape fruit, a series of changes, including color change, cell wall softening, fruit sugar accumulation, and aroma change can be observed. Among the many quality indicators of grape, the fruit skin color is the most intuitive trait. The uniform coloring of fruit can be a key factor in fruit harvesting. It is affected by many factors in the process of ripening grape including environment, hormones, climate, and cultivation conditions. Moreover, due to the high temperature and high humidity of the grape fruit ripening season, the fresh grape may be infected by *Botrytis cinerea* before and after harvesting, which is one of the most important diseases of table grape [3,4]. However, the preservation of quality and storage stability of grapes is important to consider in terms of grape consumption during the postharvest period of the fruit.

As an important plant hormone, ethylene plays an important role in plant growth and development, regulating seed germination [5], cell elongation [6], flower development [7], sex determination [8], fruit ripening [9], aging [10], and responding to both biotic and abiotic stresses [11]. The ethephon as a plant growth regulator could promote the early coloration of apples and the anthocyanin content, promote the release of ethylene, and up-regulate the anthocyanin synthesis genes [12]. It can also promote the early cracking of durian fruit [13] and promote early coloration of mango fruit and softening of fruit [14]. Exogenous ethylene treatment of mulberry fruit increased the content of soluble solids and the softening of the fruit, reduced the firmness of the fruit, and up-regulated the genes related to ethylene synthesis pathways such as *ACO1*, and *ACS1* [15]. According to whether the fruit had a peak of respiration during the ripening process, the fruit could be divided into climacteric and non-climacteric fruit. The study of exogenous ethylene in promoting the development and maturation of climacteric fruit had been studied extensively. However, the mechanism of maturation in regulating non-climacteric fruit is still unclear. A large number of studies have revealed that ABA plays an important role in the maturation process of non-respiratory climacteric fruits, such as promoting the coloration and softening of fruits of strawberry [16] and grape [2,17]. However, recent studies found that ethylene could be involved in promoting the coloration and softening of non-climacteric fruits. Moreover, Natalia [18] found that exogenous ethephon could also promote the degradation of chlorophyll in strawberry fruit, increase the content of anthocyanin and soluble sugar, and promote fruit softening through cell wall metabolic enzymes. Catharina et al. [19] found that the expression levels of Phenylalanine ammonia lyase in strawberry (*FaPAL*) and chalcone synthase in strawberry (*FaCHS*) in the transgenic strawberry fruits were significantly higher than those in the control fruits by transfecting the strawberry fruit of etr1-1(ethylene receptor gene). Although the citric acid and malic acid of the transgenic fruits were lower than the control fruits, maltose and trehalose were higher than the control fruit. A study by Giuseppe Ferrara [20] demonstrated that exogenous ethephon promoted the coloration of grape fruit, but affected the intrinsic quality of the fruit less. E. Alos [21] also found that the application of exogenous ethylene up-regulated the transcriptional levels of *ACO1*, *ACO2*, and *ACS1* in the ethylene synthesis pathway, and the transcription of ethylene sensing and signaling genes ethylene receptor gene (*ERS1a*, *ERS1b*), and CONSTITUTIVE TRIPLERESPONSE 1 (*CTR1*) through treated loquat fruit. Yin [22] used exogenous ethylene to treat green peel citrus after 150 days of flowering, and found that exogenous ethylene accelerated the degradation of chlorophyll in citrus peel, and led to the expression of chlorophyll degradation-related structural genes. Wang [23] indicated that the application of exogenous ethylene could promote the degradation of chlorophyll and the accumulation of anthocyanin in litchi fruits, and increase the concentration of 1-aminocyclopropane-1-carboxylic acid (ACC), 1-aminocyclopropane-1-carboxylic acid oxidase (ACC oxidase), and abscisic acid(ABA). Therefore, many studies confirm the important regulatory effects of ethylene on the climacteric fruit and the maturity of non-climacteric fruit ripening.

In addition, ethylene can play a critical role in the resistance to fungal pathogens such as *Botrytis cinerea*. *B. cinerea* is recognized as the second largest plant fungal pathogen in the world [24], which caused serious damage to the fruit before and after harvest. The prevention of *B. cinerea* infection has always been an important technical link in the production of fruits and vegetables. At present, there are some strategies to control the prevalence of *B. cinerea* like foliar management, fungicide application, and fumigation by sulfur dioxide after harvesting [25], but they are less considerable for consumers of fruits. Therefore, it will particularly be important to study the defense mechanism of fruits against *B. cinerea* and measure the storage stability of fruits. Some studies indicated that phytohormones such as auxin, abscisic acid, salicylic acid (SA), jasmonic acid (JA), ethylene, brassinolide, and gibberellin could be involved in plant-pathogen interactions [26,27]. Among them, JA, SA, and ethylene induced the defense mechanisms against foreign pathogenic bacteria, in which SA signaling was responsible for the induction of bionutrient and semi-nutrient pathogens, while signal transduction of JA and ET was activated against resistance to nutritional pathogens [28]. Moreover, the administration of exogenous methyl-jasmonate (MeJA) could effectively enhance the antioxidant enzyme activity in the fruit and inhibit *B. cinerea* infection of strawberry [29,30], peach [31], grape [32,33], and tomato [34,35]. Ethylene also induces resistance in plants through the expression of phytoalexin- and disease-related genes (PR) [36]. 

In this study, we investigated the role of ethylene in the development grape fruits and resistance against *B. cinerea* through the expression of resistance-related genes, ethylene metabolism pathway-related genes, and some physiological parameters. Furthermore, we evaluated the role of ethylene response factor 1 (ERF1) transcription factor as a regulatory mechanism of ripened fruits against *B. cinerea* infection in collaboration with protein ETHYLENE INSENSITIVE 3 (EIN3), which affect the storage stability of fruits against pathogenic attack.

## 2. Materials and Methods

### 2.1. Plant Materials and Treatment 

Grapevine (*Vitis vinifera*) cv. Kyoho (table grape) was used as experimental material, 8-year-old vine were collected from the experimental vineyard of Nanjing Agricultural University located at Li Shui, Nanjing, during the summer season of 2018. Fifty grape ears that were free from diseases and insect pests and have the same size as 50 bunches were collected before one week of veraison. All the collected samples were soaked into the solution of 200, 400, 600, 800 and 1000 mg/L ethephon added with 0.1% Tween 80, water was used as a control, and vacuumed three times, each time for 10 min as described by Jia [1]. The treated ears were placed in a greenhouse at a temperature of 25 °C with a relative humidity of 90%–95%. Furthermore, the samples were collected at 0, 2, 4, 6, 8, and 11 days after ethephon treatment, respectively, fruits were peeled off and immediately frozen in liquid nitrogen and stored at −80 °C until used.

### 2.2. B. cinerea Treatment

*B. cinerea* was collected from grapevines in the field and isolated in the laboratory with potato dextrose broth medium (PDB) to prepare *B. cinerea* spore suspension. Sterile scalpel was used to create a mechanical wound on the surface of the ripped grape fruit (80% maturity), and 100, 200, 300, 400, 500, 600, 700, 800, 900 and 1000 mg/L of ethephon were sprayed on the fruit surface, 24 h later 1 × 10^5^
*B. cinerea* spore was inoculated, and stored the inoculated fruit in greenhouse with a temperature of 28 °C and a relative humidity of 95%. The development of mycelium was observed on the fruit surface within 72 h with SEM. Samples (5 replicates) were collected and frozen in liquid nitrogen, and stored at −80 °C.

### 2.3. RNA Extraction and qRT-PCR Analysis

Total RNA extraction was performed using the CTAB method as described by Guan [37], genomic DNA was removed using an RNAse-free DNase I Kit (Takara, Kusatsu, Japan) according to manufacturer’s instructions. The purity of RNA was measured using a One Drop-1000+ absorbance photometer (Thermo Fisher Scientific Co. Ltd., Shanghai, China), and the RNA integrity was measured by agarose electrophoresis. The RNA sample was synthesized to the cDNA using RNA reverse transcription kit (Takara, Japan). The obtained cDNA was directly used for qPCR or storage at −20 °C.

RT-qPCR: Using *VvACTIN* as the internal reference gene, according to the SYBR Premix ExTaqTM kit (purchased from Yisheng Biotechnology Co., Ltd, Shanghai, China), the Bio-Rady-IQ2 real-time fluorescence quantitative q-PCR instrument was used to detect the relative gene expression. The reaction mixture was prepared according to the instructions of the SYBR Premix ExTaqTM kit. The amplification mixture contained 1 μL cDNA, 0.8 μL the upstream and downstream primers (Appendix A), respectively, 10 μL SYBR Premix MIX (Company, City, Country), 7.4 μL ddH_2_O, and total volume was 20 μL. The RT-qPCR procedure was pre-denaturation at 95 °C for 1 min, denaturation at 95 °C for 10 s, annealing at Tm (annealing temperature) for 20 s, extension at 72 °C for 30 s, 40 cycles, and annealing temperature of 58 °C. The experiment was set up in 3 replicates. The test data was analyzed by Excel software. The relative expression was the relative value of the treatment group and the control group by 2^−∆∆CT^ calculation. 

### 2.4. Determination of Anthocyanin, Soluble Solid, Titratable Acidity, Fruit Firmness and Falling Rate 

Grape fruit skin anthocyanin content determination was performed by pH difference method [16]. 1 g skin (5 replicates) was extracted with 1% HCl methanol and the absorbance was determined at 530 and 657 nm. The formula *A* = *A*_530_ − 0.25 *A*_657_ was used to calculate the contribution of chlorophyll and its degradation products to the absorbance at 530 nm. The anthocyanin concentration was a relative value, and the calibration were set as *A* = 0.01 equal to 1 unit. The content of soluble solids (TSS) was measured using a portable digital hand-held dialyzer (PAL-1, ATAGO Co. Ltd., Tokyo, Japan), the titratable acidity was titrated with 0.1 mol/L NaOH. The firmness was measured using a hand-held firmness tester (GY-2). The calculation formula of falling rate was: (falling grape fruit/total grape fruit) × 100% (10 replicates)

### 2.5. Determination of Antioxidant Enzyme Activity

Catalase(CAT)and Superoxide dismutase(SOD)activities were measured as described by Zhang [38]. 5 g frozen flesh (5 replicates) was homogenized in 50 mM phosphate buffer (pH 7.8) containing 0.2 mM EDTA and 2% polyvinyl pyrrolidone (PVP). The homogenate was centrifuged at 12,000× *g* for 20 min at 4 °C, and the supernatant was used for CAT, and SOD activities determination. One unit of CAT activity was defined as a decrease in absorbance at 240 nm of 0.01 per min. One unit of SOD activity was defined as an enzyme that caused a 50% inhibition of nitro blue tetrazolium (NBT) reduction under assay conditions.

Polyphenol oxidase(PPO) enzyme activity was assayed by the method described by Rastegar [39], with some modifications. The enzymes were extracted by homogenizing 0.2 g of frozen flesh (5 replicates) in phosphate buffer solution (pH 7 and 4% polyvinylpolypyrrolidone). After centrifugation at 16,000× *g* for 30 min at 4 °C (Centrifuge Hettich rotofix 32, Tuttlingen, Germany). A volume of 1800 μL of phosphate buffer solution with pH 7 (0.1 M), 600 μL of catechol (0.1 M) and 600 μL of the enzyme extract were mixed. The increase in absorbance was read at 410 nm for 3 min and the results were expressed as U mg^−1^ FW.

Total flavonoids content was quantified as described by Chang [40]. A quantity of 1 g fruit flesh (5 replicates) was added to 0.5 mL methanol, and then 0.1 mL of 10% AlCl_3_ and 0.1 mL of 1 mmol/L acetate potassium solution were added to incubate. After 30 min, the absorbance was read at 415 nm using a UV–vis spectrophotometer (Cecil Instrumentation Services Ltd., Cambridge, England). Quercetin was used as a standard for the construction of the calibration curve. The results expressed as mg of quercetin equivalents per g of FW.

### 2.6. Construction of Overexpression or RNA Interference Vector and Agrobacterium-Mediated Infiltration

For overexpression of VvERF1 gene, the 672 bp cDNA of VvERF1 was amplified by using primers (sense 5′-AGATCTATGGATTCTTCTTCCTTCTA-3′, antisense 5′-ACTAGTTGATGAACACAAGAGTTGCT-3′), and forward inserted into pCAMBIA1302 using Bgl Ⅱand Spe Ⅰdigestion site underlined. For RNA interference of gene VvERF1, the 672 bp cDNA of VvERF1 was amplified using primers (sense 5′-ACTAGTATGGATTCTTCTTCCTTCTA-3′, antisense 5′-AGATCTTGATGAACACAAGAGTTGCT-3′), and reverse inserted into pCAMBIA1302 using Spe Ⅰ and Bgl Ⅱ digestion site. pCAMBIA1302 or these pCAMBIA1302 derivatives were transformed into Agrobacterium strain EH105 by the freeze-thaw method. For each strain, the 5 mL culture was grown overnight at 28 °C in Luria-Bertani (LB) medium (50 mg mL^−1^ kanamycin and 50 mg mL^−1^ rifampicin, 10 mM MES, 20 μm acetosyringone). The overnight cultures were inoculated into 50 mL of LB medium and grown at 28 °C overnight. The cells were harvested by centrifugation (5000 rpm, 5 min, 20 °C), resuspended in infiltration buffer (10 mM MgCl_2_, 10 m MMES, 20 μm acetosyringone), adjusted to an optical density (OD_600_) of 1.0–2.0, and left to stand at room temperature for 4 h. About 1 mL of Agrobacterium was infiltrated into every strawberry fruit (12 days after flowering) with a 1 mL syringe. Ten uniformly sized fruits were used in the infiltration experiment, and the experiment was repeated three times. Twenty-four hours after injection, *B. cinerea* was inoculated at the same position on the surface of the strawberry fruit. The number of diseased strawberries and the diameter of the lesions were measured 2 days, 4 days and 6 days after inoculation.

### 2.7. Scanning Electron Microscopy (SEM) Observation

SEM analysis was conducted as described by Qi [41]. Mycelium were collected from the surface of the fruit and the glutaraldehyde/paraformaldehyde-fixed mycelium was post-fixed by 2% osmium tetraoxide (OsO4) and the final fixation step was performed overnight by 2% tannic acid/guanidine hydrochloride. It was rinsed 3 times with PBS buffer at 4 °C for 15 min. After that, the mycelium was dehydrated in continuous ethanol: 0%, 50%, 70%, 85%, 95% once, and 100% twice (15 min each time). Mycelium samples were dried in a vacuum freeze dryer and used Ion sputtering coating machine (E1045, Hitachi, Tokyo, Japan). Painted samples were used for scanning electron microscope SEM (TM3000, Hitachi, Tokyo, Japan).

### 2.8. Yeast Two-Hybrid(Y2H) Assay

The Matchmaker GAL4 Two-Hybrid Systems were used for Yeast two-hybrid assays. To validate the interaction of VvERF1 with VvEIN3, the full-length cDNA sequences of VvERF1 and VvEIN3 were subcloned into pGBKT7 and pGADT7 vectors, respectively. Different combinations of constructs were co-transformed into yeast strain Y2H Gold by the lithium acetate method, and yeast cells were cultured on minimal medium/-Leu -Trp. Several single colonies grown on minimal medium were picked and inoculated into minimal medium containing 20 μg mL^−1^ 5-bromo-4-chloro-3-indolyl-α-d-galactopyranoside/-Leu/-Trp/-His/-adenine to test for possible interactions. Primers for the yeast two-hybrid assay are listed in Appendix A.

### 2.9. Statistical Analysis

SPSS Statistics software was used for statistical analysis for recorded data and was conducted in triplicate, significant differences were observed at *p* values of less than 0.05. Figures were expressed as mean ± standard error. Comparison of means was performed by using Duncan’s multiple range tests [42]. Statistical analysis of the obtained data was performed using SPSS Statistics software (IBM^®^SPSS^®^Statistics, New York, NY, USA).

## 3. Results

### 3.1. Effect of Ethephon on Grape Coloring

To investigate the effect of different concentrations of ethephon on grape fruit coloring and seed fall, grape ears were treated with 200, 400, 600, 800, and 1000 mg/L of ethephon one week before veraison. Results showed that ethephon promoted the coloring of grape fruits at 600 mg/L after 6 d (Figure 1A). The grape colored index and the coloring rate of grape fruits were also increased after 6 d with the 600 mg/L ethephon treatment (Figure 1B). Interestingly, ethephon at concentrations above 600 mg/L did not significantly increase the coloration coefficient of grape fruit, but promoted the falling of the fruit and shrunk the fruit (Figure 1C). Furthermore, the anthocyanin content of grape fruits reached a peak (29.69 mg/kg) at 6 d when the sample were treated with 600 mg/L ethephon, while the anthocyanin content of the control group was only 22.36 mg/kg (Figure 1D). Among them, the treated samples with 1000 mg/L of ethephon indicated a falling rate of 28% at 6 days and a detachment rate of 60% at 11 days, which severely affected the fruit quality. In general, we found that ethephon could significantly promote the coloration of grape fruits.

### 3.2. Effects of Ethephon on Grape Fruit Quality

To investigate the effect of ethephon on grape fruit quality, the weight loss rate, firmness, soluble solids, and titratable acidity of ‘Kyoho’ grape fruit at 0, 2, 4, 6, 8, and 11 days after ethephon treatment were measured. Results showed that different concentrations of ethephon reduced fruit firmness, especially at 800 mg/L and 1000 mg/L of ethephon treatment (Figure 2A). The highest concentration of ethephon (1000 mg/L) significantly decreased fruit firmness that was 3.8125 kg/cm^2^ at 11 days, however, the firmness of the control group was 7.215 kg/cm^2^. Meanwhile, the changes of soluble solids and titratable acidity were not changed significantly after ethephon treatment (Figure 2B,C). Therefore, ethephon as a plant growth regulator had not affected the intrinsic quality index of the fruit. The weight loss rate showed an upward trend after the application of ethephon, and the effect was most obvious after the 1000 mg/L ethephon treatment (Figure 2D).

### 3.3. Effects of Ethephon on Grape Cell Wall Metabolism-Related Enzyme Activities.

The softening level of the fruit is a critical index during storage that some important enzymes like PG, PME, PE, and cellulase affect fruit softening and cell wall metabolism. PG activity as one of the key enzymes of fruit ripening and softening increased after ethephon treatment at 11 d, although its activity was slow until day 4 (Figure 3A). In addition, PG enzyme activity in the control was lower than ethephon treatment. Meanwhile, the enzyme activity of PME raised after ethephone treatment and reached a peak at 8 and 11 d (Figure 3B). Different concentrations of ethephon increased PE activity and reached its peak at 6 d but then decreased until day 11 (Figure 3C). Similar to PME pattern, cellulase activity also showed a continuous upward trend after ethephon treatment, reaching its maximum at day 11 (Figure 3D).

### 3.4. Effects of Ethephon on Grape Fruit Maturation and Hormonal Metabolism-Related Genes

#### 3.4.1. Anthocyanin Anabolic Pathway Genes

Consistent with the grape fruit anthocyanin content, exogenous ethephon increased the expression levels of some regulatory genes like *MYBA1* and *MYBA2*, and grape fruit anthocyanin synthesis related genes like CHS, CHI, F3H, F3’HF3’5’H, UFGT and GST in a dose-dependent manner when compared to control (CK) (Figure 4A–I). Regulatory genes in the anabolic pathway of anthocyanin (*VvMYBA 1*, *VvMYBA 2)* and anthocyanin synthesis genes *VvF3’H*, *VvGST* firstly indicated an increase in their expression levels increased and then decreased after ethephone treatment in different concentrations, while the expression levels of other genes related to anthocyanin synthesis pathway remained unchangeable after increasing (Figure 4).

#### 3.4.2. Cell Wall Metabolism and Aroma Metabolism Genes

After treatment with ethephon, softening-related genes like *VvPG*, *VvPL*, *VvPME*, and *VvCEll* were up-regulated compared to control, however, the expression levels were changeable in different concentrations of ethephon (Figure 5A–D). At the higher concentration of ethephon, higher levels of gene expression were observed. After 6 d, the 1000 mg/L ethephon treatment group promoted the expression of *VvPG* and *VvPME* genes, while 600 mg/L ethephon treatment induced *VvPL* and *VvCEll*. In general, 600 mg/L treatment of ethephon induced the expression of fruit softening genes more than other treatments at 6 d. Three genes of the grape aroma anabolic pathway including *VvQR*, *VvEcar*, and *VvEGS* [2] showed different expression patterns, which the expression level of *VvQR* had been increasing during processing; however, there were differences between different concentrations (Figure 5E). Meanwhile, the expression pattern of *VvEcar* was oscillatory, while *VvEGS* decreased until day 4, but then increased until day 11 (Figure 5F,G).

#### 3.4.3. ABA and Ethylene Synthesis Pathways

Different concentrations of ethephon treatment affected the expression of abscisic acid and ethylene metabolic pathway genes. In the abscisic acid (ABA) synthesis pathway, 9-cis-epoxy carotenoid dioxygenase (NCED) is a key rate-limiting enzyme with different isoforms in Vitis including *VvNCED1*, *VvNCED2*, and *VvNCED3* [43]. The expression patterns of these three genes changed in a concentration-dependent manner after ethephon treatment. Among them, the expression of *VvNCED1* was unchangeable until day 2, but it increased from day 4 to day 11 (Figure 6A). The expression levels of *VvNCED2* and *VvNCED3* did not change significantly during the first 4 d, but increased at 6 d and reached the highest peak at 11 d (Figure 6B,C). Β-glucosidase (BG) is also an important enzyme for ABA synthesis, which we measured the different isoforms including *VvBG1*, *VvBG2*, and *VvBG3*. *VvBG1* showed a rising trend during time course experiments (Figure 6D), however, the expression level was relatively low in the first 4 d and continued to reach the maximum level at 11 d. Meanwhile, the expression level of *VvBG2* significantly increased at day 6 after ethephon treatment and the expression trend remained approximately unchangeable until day 11 (Figure 6E). The expression pattern of *VvBG3* was different and firstly increased and then decreased, although there were differences between different ethephon treatments (Figure 6F). Ethephon treatment also inhibited the expression level of the ABA inhibitory gene *VvCYP7071* compared to control (Figure 6G). The expression patterns of the ethylene synthesis and anabolic pathway genes like *VvACS1*, *VvETR2*, *VvEIN3*, and *VvACO2* at different concentrations of ethephon were slightly different. Among them, the expression level of *VvACS1* was low at day 6 and raised at day 8 (Figure 6H). However, the expression level of *VvETR2* (Figure 6I) at 11 d was not obvious, and the expression level of *VvEIN3* (Figure 6J) first raised slowly, then significantly increased at day 8 and 11 compared to day 6. Meanwhile, expression level of *VvACO2* showed a trend of increase in a concentration-dependent manner of ethephon treatment (Figure 6K).

#### 3.4.4. Ethephon Stimulates Brassinolide Pathway and Inhibits Auxin Synthesis

We investigated some key genes involved in brassinolide synthesis pathway and auxin synthesis to evaluate the crosstalk between growth regulators in grape fruits. Results indicated that some genes of the auxin anabolic pathway including Auxin transporter gene *VvPIN*, synthesis gene indole synthase (INS), indole-3-pyruvate monooxygenase YUCCA (YUC), and tryptophan aminotransferase of Arabidopsis1 (TAA1) were inhibited with increasing concentration of ethephon (Figure 7A–D). However, the treated samples with different concentrations of ethephon indicated a significant increase in the expression levels of genes related to the brassinolide synthesis pathway *VvBR60X*, *VvDWF*1 (Figure 7E,F). Therefore, ethephon could suppress and induce auxin synthesis and brassinolide, respectively through down-regulation and up-regulation of hormonal responsive genes.

#### 3.4.5. Ethephon Activates Jasmonate Signaling

To understand the relationship between ethephon and jasmonate, we evaluated the expression levels of genes related to the jasmonic acid anabolic pathway such as *VvJAZ9*, *VvJAZ*4, *VvCOZ*1, *VvAOS*, and *VvLOX*. The effects of different concentrations of ethephon treatment on grape fruit increased transcript levels of five genes, which are necessary for jasmonate signaling (Figure 8A–E). Therefore, ethphon could also activate jasmonate signaling through two key genes of JA biosynthesis including *VvAOS*, and *VvLOX* and mediated downstream processes through binding of JAZ proteins to the F-box protein CORONATINE INSENSITIVE1 (COI1), part of the Skp1/Cullin/F-box SCF^COI1^ ubiquitin E3 ligase complex.

### 3.5. Ethephon Increases Resistance to B. cinerea Infection in Grape Fruits

To evaluate the ethephon effect in pathogenic attack of fruit, samples were treated with *B. cinerea* after ethephon pretreatment and we monitored the damage rate of grape fruits. Results showed that the different concentrations of ethephon on grape fruit significantly decreased botrytis infection (Figure 9A) compared to control during the time course experiments. Within 72 h of ethephon treatment, the growth of *B. cinerea* mycelium on the surface of grape fruit treated with low-concentrations of ethephon (100, 200, 300, 400 mg/L) was significantly higher than treated samples with the high-concentration of ethephon. As shown in Figure 9, the incidence rate of the ethephon treatment group significantly decreased in a concentration-dependent manner of ethephon, although the low-dose of ethephon indicated inhibitory effects less on growth *B. cinerea* on the surface of grape fruit. Furthermore, we collected the *B. cinerea* hyphae on the surface of grapes treated with 1000 mg/L ethephon at 72 h and compared to control by electron microscopy (Figure 9B). The mycelium structure on the surface of grape fruits of the control group was relatively complete, while *B. cinerea* mycelium had shrunk in pretreated samples with ethephon. We also calculated lesion diameter and disease incidence of botrytis infection after treatment with ethephon, and our finding confirmed the inhibition of botrytis infection by ethephon, especially in high concentrations (Figure 9C).

### 3.6. Ethephon Activates Antioxidant System in Fruits after Botrytis Inoculation

We considered the effect of ethephon and fungal attack on the oxidative stress of grape fruits and identified three antioxidant key enzymes including superoxide dismutase (SOD), polyphenol oxidase (PPO), and catalase (CAT) in grape fruits. Our finding indicated that ethephon pretreatment could increase the activities of three enzymes SOD, PPO, and CAT after botrytis infection (Figure 10A,C,D).

After treatment with ethephon, the activity of PPO increased with the increase of the ethephon concentration (Figure 10A). The change of SOD content was shown at the concentration of 400 mg/L ethephon, the overall trend was increased, while the concentration of ethephon above 400 mg/L reached the highest value of activity at 1 h, and decreased within 48 h and 72 h (Figure 10D). The change of CAT activity generally showed an upward trend with the increase of ethephon concentration. The high concentration of ethephon promoted the activity of CAT in grapes to enhance its resistance to exogenous *B. cinerea* (Figure 10C). Furthermore, total flavonoids as antioxidant active substances were accumulated in grape fruits after ethephon treatment inoculated with *B. cinerea*. However, firstly the total flavonoid content increased and then decreased with the increase of ethephon concentration (Figure 10B).

### 3.7. Effect of Ethephon Treatment on the Expression of Disease Resistance Genes in Grape Fruits after B. cinerea Inoculation

To investigate the effect of ethephon treatment on the expression of disease resistance-related genes in grape fruit, we inoculated grapes with *B. cinerea* after the ethephon pretreatment and monitored at 1 h, 48 h, and 72 h. The expression levels of *VvPIP1*, *VvNAC26*, *VvDREB*, *VvAPX*, *Vvpgip*, *VvWRKY70*, *VvMYC2*, *VvNAC*, and *VvPAD4* were measured (Figure 11A–I). We found that the expression levels of *VvNPR1* and *VvPAD 4* increased within 72 h after ethephon pretreatment, which correlated with the increase of the ethephon concentration and an enhancement of resistance to *B. cinerea* infection. The lower concentration of ethephon affected less the transcript level of *VvWRKY70* and gradually increased from 400 mg/L and reached the maximum level of expression at 72 h. However, the other genes did not show a clear pattern of expression (Figure 11).

### 3.8. Overexpression and Interference of VvERF1 Response to B. cinerea

As an important transcription factor, ethylene response factor (ERF) transcription factor plays an important role in ethylene signaling. We cloned the grape *VvERF1* gene and constructed an overexpression and interference vector for transient expression in strawberry fruits. Results showed that, on the second day, only the fruit surface of VvERF1-RNAi had *B. cinerea* hyphae, and there was basically no change between VvERF1-OE and control (Figure 12A). We observed that transformed strawberry fruits with VvERF1-OE were resistant to *B. cinerea* within 6 d and could inhibit the botrytis infection compared to control and VvERF1-RNAi (Figure 12A). Furthermore, the diameter of the lesions on the surface of the fruit and the disease incidence were quantified. We observed that the lesion diameter of VvERF1-RNAi and control reached 2.6 cm and 2.3 cm respectively, while it significantly decreased by VvERF1-OE (0.8 cm) (Figure 12B). Our finding also indicated that the disease incidence rate of VvERF1-RNAi and control group reached maximum levels (100%) after 6 d of *B. cinerea* infection, while the disease incidence of VvERF1-OE was less (85%) (Figure 12C). The fungal biomass on the surface of VvERF1-OE strawberry fruits was less than the control at 6 d, but VvERF1-OE indicated more than the control (Table 1). Therefore, we suggest that overexpression of VvERF1 lead to resistance against *B. cinerea* in ripened fruits.

### 3.9. VvERF1 Interacted with VvEIN3

To verify whether *VvERF1* and *VvEIN 3* interacted with each other and played a role in ethylene regulation of grape maturity, we used yeast two-hybrid to verify the interaction between the two proteins. We observed that AD-VvERF1 plus BD-VvEIN3, BD-VvERF1 plus AD-VvEIN3, BD-SV40 plus AD-P53 increased the activity of β-galactosidase compared to negative controls (Figure 13A,B). Therefore, it confirmed that VvERF1 interacted with VvEIN3. Furthermore, the grape fruit epidermis was treated with ethylene and inhibitors of ethylene biosynthesis like aminoetoxyvinylglycine (AVG) and aminoxyacetic acid (AOA), and the expression levels of VvERF1 and VvEIN3 were determined after 3 d. We found that the expression of VvERF1 and VvEIN3 significantly increased in response to ethylene, while the suppression of ethylene biosynthesis decreased their expression levels (Figure 13C).

## 4. Discussion

### 4.1. Influence of Ethylene on the Grape Fruit Physiological and Molecular Changes

The growth and development of plants are associated with the phytohormones, transcription regulators and mechanical properties. Fruits produce a gaseous compound called ethylene, which act as an important hormone in fruits ripening, maturation and disease resistance. Which is evident from the various past research findings in climacteric fruits [44]. Although relatively few studies have been carried out on non-climatic fruits, recent studies have shown that ethylene also has an important role to play in the coloring and maturation of grape berries [45,46]. However, the application of ethephon observed dual effect on grape fruit ripening. The effect of ethylene on the ripening of grape fruit is different due to the different development stages of the grape fruit. The application of ethephon at the early stage of grape fruit development can delay the ripening of the grape fruit, while the application of ethephon before veraison stage of the grape can promote the fruit ripening. The reason for this effect may be related to the change of auxin content. The early application of exogenous ethylene can promote the accumulation of auxin in the grape fruit, and there will be a short peak of ethylene release in the grape fruit before veraison stage [47,48,49]. In our study, we used exogenous ethephon to treat the grape fruit (7 days before veraison stage) and observed that the application at a certain concentration of ethephon promoted the color change of the grape fruit; however, the high concentration (1000 mg/L) inhibited the coloration of the grape fruit and also induced abortion of grape fruit fall. After 6 d of treatment, 600 mg/L ethephon encouraged early color change and increased the color coefficient and anthocyanin content of grape barriers. Meanwhile, various concentrations of ethephon had little effect on the intrinsic quality of grape fruits (soluble solids, titratable acid content). Previous studies have also confirmed that the treatment of grape fruits with ethephon can improve the fruit skin color and reduce the firmness, but has little effect on the intrinsic quality indicators [20,50]. Ethephon treatment of uncolored fruits could promote chlorophyll degradation and anthocyanin synthesis related genes [51]. By increasing the expression of anthocyanin-related genes to increase the anthocyanins content, in the present study we found that ethephon treatment promoted *VvCHS*, *VvUFGT*, *VvCHI*, *VvF3H*, *VvF3’H* and *VvF3’5’H* expression in grape fruit peel. The expression of important structural genes such as the anthocyanin synthesis pathway increased the biosynthesis of anthocyanin content. This was consistent with the previous study using 2-CEPA (ethylene release compound) to treat grape berries can positively regulate the expression of anthocyanin synthesis-related genes, like *CHS*, *F3H*, and *UFGT* [52]. Exogenous ethylene also played an important role in the softening process of climacteric fruits. Previous studies confirmed that the application of exogenous ethylene could reduce the firmness in apple and tomato fruits; however, it also promotes the expression of the *XTH* gene, and accelerates the degradation of cell walls to promote fruit softening [47]. Relatively few studies have been published in context of non- climacteric fruits. In our study, we found that the application of exogenous ethephon also increased grape fruit softening and the activity of enzymes related to cell wall metabolism (PG, PE, PME, Cellulase). Meanwhile, ethephon also promoted the expression levels of fruits ripening and softening-related genes (*VvPG*, *VvPL*, *VvPME*, *VvCell*), and difference was evident between different levels of concentrations. As the concentration increased, hormone activity was a major factor influencing fruit maturity. Ethephon application could regulate gene expression levels associated with other hormone biosynthesis pathways. It was well known that abscisic acid played an important role in the maturation of non- climacteric fruits, which could promote the early color conversion of immature fruits and the sugar content accumulation [2]. In this study we found that the application of ethephon increased the expression of *VvNCED1*, *VvNCED*2, and *VvNCED3*, which are important genes in the abscisic acid synthesis pathway. Furthermore, the expression levels of *VvBG1*, *VvBG2*, and *VvBG3* were also increased. Grape is a non- climacteric fruit and has a peak ethylene release during the ripening process, but after treatment with ethephon, it also slightly promoted expression of *VvACS1*, *VvETR2*, *VvEIN*3 and *VvACO2* in ethylene biosynthesis pathways. This was consistent with previous results on non-climacteric fruits [48]. We also found that ethephon promoted the expression of genes related to the brassinolide synthesis pathway and jasmonate synthesis pathway, and inhibited genes expression related to the auxin synthesis pathway.

### 4.2. Effects of Ethephon on Grape Fruit Resistance to B. cinerea

The application of synthetic biomolecules has great impact on plant resistance. Among them, the roles of SA, JA and ET and synergistic and antagonistic interactions in plant diseases and immune responses had been validated in a variety of plants [53]. Exogenous application of methyl jasmonate could improve the resistance of Chilean strawberry to *B. cinerea*, and promote the expression level of disease-related genes such as *FaPGIP* and *FaCHI* [29]. Application of exogenous methyl jasmonate to grape fruit could inhibit the growth of *B. cinerea*, and increase fruit disease-related enzymatic activities and disease resistance in fruits [33]. Jia also confirmed that methyl jasmonate had a positive role in the resistance of grape fruits to against *B. cinereal* [1]. Salicylic acid and ethephon treatment were carried out on tomato fruits at different stages of maturity, the variable growth stages showed different resistance to *B. cinerea*, however all treatments inhibited the occurrence of *B. cinerea*, and the incidence of *B. cinerea* and the diameter of lesions were reduced. It also increased the expression of disease-related proteins like PR1 and PR 3 and improved resistance against diseases resistance [54]. Application of 1 μL/L ethylene to tomato plants could significantly reduce the incidence of *B. cinerea* disease per plant. However, it could strengthen genes in terms of disease resistance. Ethylene can also participate in the disease resistance of plants with other substances. Overexpression of yeast spermidine synthase (ySpdSyn), an enzyme involved in polyamine (PA) enhances the susceptibility of tomato to *Botrytis cinerea*, while the ethylene precursors ACC and SAM are used for inoculation after the spores of *Botrytis cinerea*, the disease resistance of the transgenic tomato was enhanced [55]. However, relatively few studies have been conducted on the direct application of ethylene to the fruit to explore disease resistance. In our study, *B. cinerea* inoculated after ethephon treatment on mature grape fruits and found that after different concentrations of ethephon showed variable results in disease resistance to grape barriers. Meanwhile after 72 h inoculation, we found that low-level ethephon prolonged time period had a poor defense against *B. cinerea*. In addition, as the concentration levels were increased, it inhibited the growth of *B. cinereal* and diameter of the lesion. The inhibition of incidence of disease and the diameter of the lesion was obvious after ethephon treatment. Ethylene can be used as a resistance regulator to improve the disease resistance of the fruit, enhance the antioxidant enzymes in the fruit and upregulate the expression of disease-related genes.

### 4.3. Ethylene Response Factor 1 (ERF 1) Played an Important Role in the Resistance of Grape Fruits to B. cinerea

Ethylene response factor (ERF) is an important member of the AP2/ERF family and played an important role in resistance against biotic and abiotic stresses [56]. Among them, it played a significant role in resisting exogenous pathogenic bacteria. VaERF20 improved the resistance to *B. cinerea* in transgenic Arabidopsis and tomato, and increased the expression of disease resistance genes such as *AtPR1*, *AtLOX3* and *AtPDF1.2* in transgenic plants [57]. In Arabidopsis, ERF5 and ERF6 played active roles in shielding against exogenous *B. cinerea*, and reduced the disease index, with increase in the expression of disease-related genes [58]. Overexpression of AtERF15 had anti-grey mold properties in Pseudomonas [1] syringae pv.tomato DC3000, which inhibited the growth of *B. cinerea*, and interfered with AtERF15 reduced the inhibitory effect of *B. cinerea* [59]. Previous studies also confirmed MPK3/MPK6’s phosphate against ERF6, phosphorylated ERF6 could constitutively activate defense-related genes, especially genes related to fungal resistance, including *PDF1.1* and *PDF1.2*, and enhanced resistance to *B. cinerea* [60]. In tomato, overexpression of SlERF2 reduced the incidence of *B. cinerea* and the diameter of lesions, while increasing the activity of disease-resistant enzymes CHI, GLU, PAL, and POD, and increasing the content of disease-related proteins PR1 and total phenol, improving disease resistance [61]. ERF1 played a positive role in defending against exogenous pathogenic bacteria. Overexpression of ERF1 gene improved the disease resistance of transgenic Arabidopsis plants, and up-regulated the expression of *PDF2.1* and *CHI* [62]. Similarly, research by Marta et al. also confirmed development of an over-expressing ERF1 vector could improve the expression of resistance-related genes. Overexpression of ERF1 gene was able to increase resistance to fusarium wilt in cucumber. It could be seen that ERF transcription factors played a positive role in plant defense against pathogen infection. In this study, constructing a VvERF-OE overexpression vector and a VvERF1-RNAi interference vector to transiently transform the results to strawberry fruits, it was observed that overexpression of VvERF1 inhibited the growth of hyphal and suppressed the germination rate and increased disease incidence rate of *B. cinerea* spores on the surface of strawberry fruits. Interference with VvERF1 promoted the growth of *B. cinerea* mycelium, and increased spore germination and disease incidence. These results showed that VvERF1 plays important role resistant against *B. cinerea* infection downstream of ethylene.

## 5. Conclusions

The color of grape fruits was affected by exogenous ethephon. Ethephon significantly increased anthocyanin content through up-regulation anthocyanin synthesis pathway genes. A high concentration of ethephon could also promote fruit shattering, raise the enzymes related to cell wall metabolism of grape fruit, and lead to the promotion of grape fruit softening. Ethephon pretreatment also increased resistance in ripened fruits in a dose-dependent manner when samples were inoculated with *B. cinerea*. In addition, it led to inhibition of the growth of *B. cinerea* by ethylene signaling of overexpressed the ethylene response factor 1(ERF1) transcription factor in collaboration with protein ETHYLENE INSENSITIVE 3 (EIN3). Therefore, we suggest that fruits pretreated with ethephon can be supported against pathogenic attacks, preserved during post-harvest processing, and increased storage stability of mature fruits.

## Figures and Tables

**Figure 1 foods-09-00892-f001:**
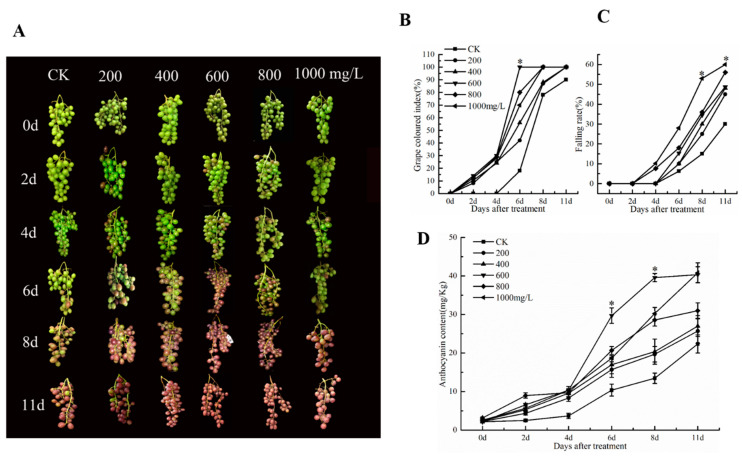
Effect of ethephon on coloring and threshing in grape. (**A**) Morphology of grape clusters. (**B**) Grape colored index. (**C**) Falling rate of grape fruit. (**D**) Changes of anthocyanin content in grape. Ethephon was used in different concentrations (200, 400, 600, 800, and 1000 mg/L) and in a time course experiments (2, 4, 6, 8, and 11 days). Values are means ± SD of five biological replicates. * Significant differences compared with the control (water-treated fruits) at *p* < 0.05, using Student’s test. d: days.

**Figure 2 foods-09-00892-f002:**
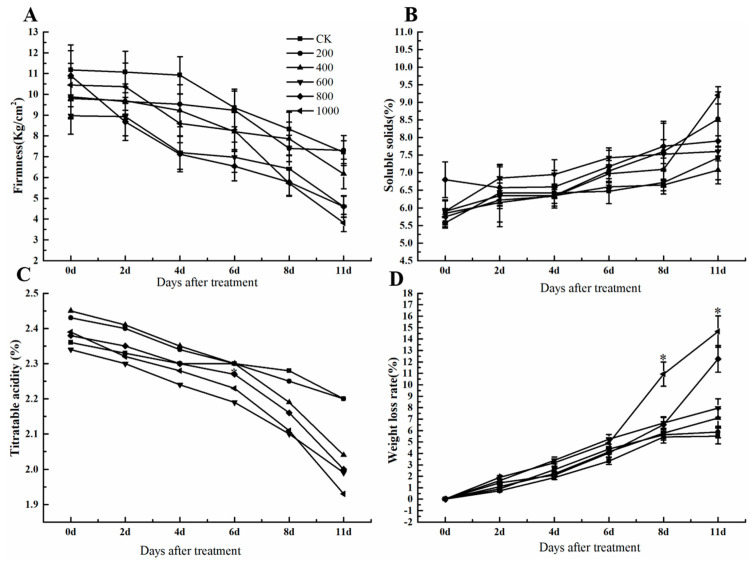
Effect of ethephon on soluble solids, titratable acidity, firmness, and weight loss rate in grape. (**A**) Fruit firmness. (**B**) Fruit soluble solids. (**C**) Titratable acidity. (**D**) Weight loss. Ethephon was used in different concentrations (200, 400, 600, 800, and 1000 mg/L) and in a time course experiments (2, 4, 6, 8, and 11 days). Values are means ± SD of five biological replicates. * Significant differences compared with the control (water-treated fruits) sample at *p* < 0.05.

**Figure 3 foods-09-00892-f003:**
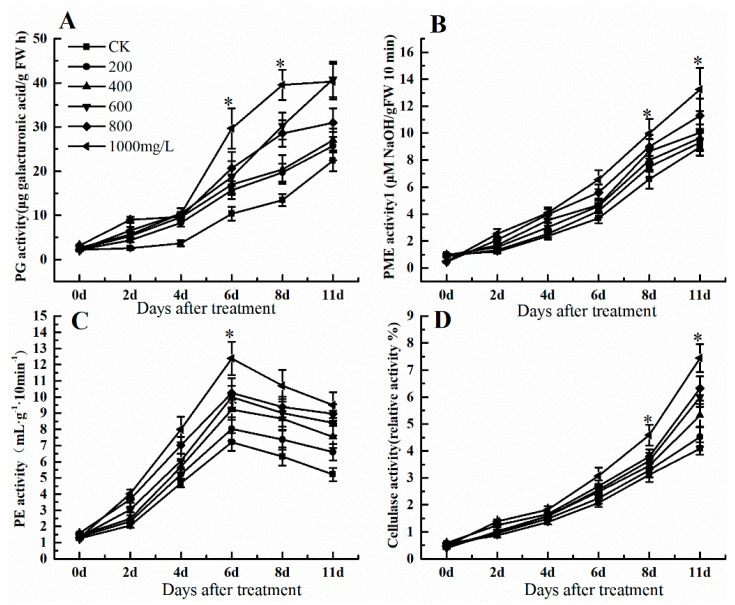
Effect of ethephon on cell wall metabolism-related enzyme activities in grape. (**A**) Polygalacturonase (PG) activity. (**B**) Pectin methylesterase (PME) activity. (**C**) Pectinesterase (PE) activity. (**D**) Cellulase activity. Ethephon was used in different concentrations (200, 400, 600, 800, and 1000 mg/L) and in a time course experiments (2, 4, 6, 8, and 11 days). Values are means ± SD of five biological replicates. * Significant differences compared with the control (water-treated fruits) sample at *p* < 0.05, using Student’s test.

**Figure 4 foods-09-00892-f004:**
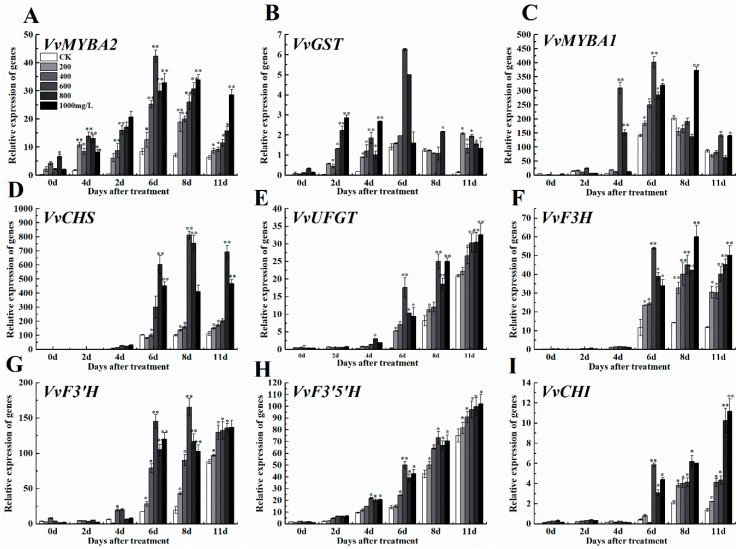
Effect of ethephon on anthocyanin-related genes expression in grape. (**A**) *VvMYBA2*, (**B**) *VvGST*, (**C**) *VvMYBA1*, (**D**) *VvCHS*, (**E**) *VvUFGT*, (**F**) *VvF3H*, (**G**) *VvF3’H*, (**H**) *VvF3’5’H*, (**I**) *VvCHI*. Ethephon was used in different concentrations (200, 400, 600, 800, and 1000 mg/L) and in a time course experiments (2, 4, 6, 8, and 11 days). Values are means ± SD of three biological replicates. * Significant differences compared with the control (water-treated fruits) sample at *p* < 0.05, ** Significant differences compared with the control (water-treated fruits) sample at *p* < 0.01 using Student’s test.

**Figure 5 foods-09-00892-f005:**
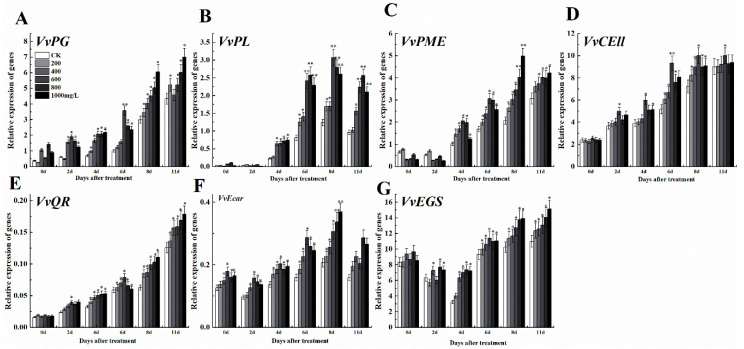
Effect of ethephon on ripening-related genes expression in grape. The expression levels of softening-related genes (**A**) *VvPG*, (**B**) *VvPL*, (**C**) *VvPME*, and (D) *VvCELL*. Aroma synthesis-related gene expression levels of (**E**) *VvQR*, (**F**) *VvEcar*, and (**G**) *VvEGS*. Ethephon was used in different concentrations (200, 400, 600, 800, and 1000 mg/L) and in a time course experiments (2, 4, 6, 8, and 11 d). Values are means ± SD of three biological replicates. * Significant differences compared with the control (water-treated fruits) sample at *p* < 0.05, ** Significant differences compared with the control (water-treated fruits) sample at *p* < 0.01 using Student’s test.

**Figure 6 foods-09-00892-f006:**
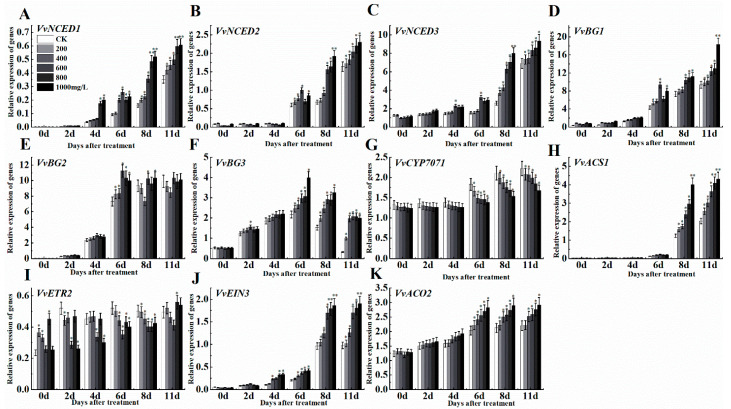
Effect of ethephon on ABA and ethylene-related genes expression in grape. ABA metabolism-related gene expression levels of (**A**) *VvNCED1*, (**B**) *VvNCED2*, (**C**) *VvNCED3*, (**D**) *VvBG1*, (**E**) *VvBG2*, (**F**) *VvBG3*, (**G**) *VvCYP707A*. Ethylene metabolism-related gene expression levels of (**H**) *VvACS1*, (**I**) *VvETR2*, (**J**) *VvEIN3* and (**K**) *VvACO2*. Ethephon was used in different concentrations (200, 400, 600, 800, and 1000 mg/L) and in a time course experiments (2, 4, 6, 8, and 11 days). Values are means ± SD of five biological replicates. * Significant differences compared with the control (water-treated fruits) sample at *p* < 0.05, ** Significant differences compared with the control (water-treated fruits) sample at *p* < 0.01 using Student’s test.

**Figure 7 foods-09-00892-f007:**
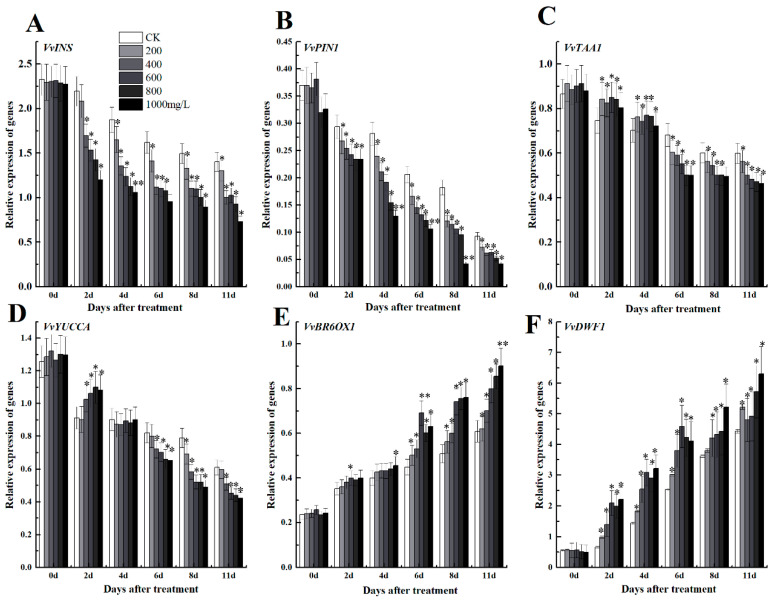
Effect of ethephon on auxin metabolism-related gene expression levels of (**A**) *VvINS1*, (**B**) *VvPIN1*, (**C**) *VvTAA1* and (**D**) *VvYUCCA*. Brassinolide metabolism-related gene expression levels of *(***E**) *VvBR6OX1* and (**F**) *VvDWF1*. Ethephon was used in different concentrations (200, 400, 600, 800, and 1000 mg/L) and in time course experiments (2, 4, 6, 8, and 11 days). Values are means ± SD of five biological replicates. * Significant differences compared with the control (water-treated fruits) sample at *p* < 0.05, ** Significant differences compared with the control (water-treated fruits) sample at *p* < 0.01 using Student’s test.

**Figure 8 foods-09-00892-f008:**
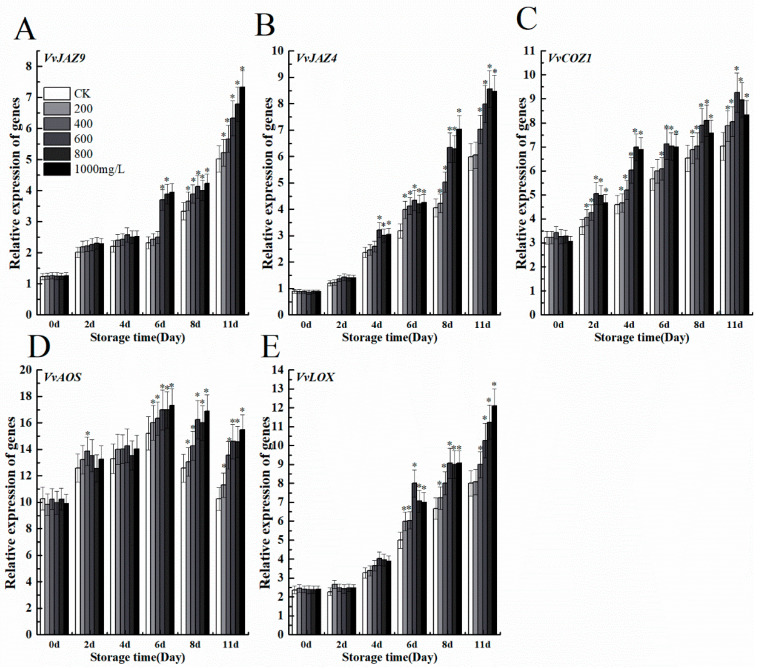
Effect of ethephon on JA-related genes expression in grape. JA metabolism-related gene expression levels of (**A**) *VvJAZ9*, (**B**) *VvJAZ4*, (**C**) *VvCOZ1*, (**D**) *VvAOS*, and (**E**) *VvLOX*. Ethephon was used in different concentrations (200, 400, 600, 800, and 1000 mg/L) and in time course experiments (2, 4, 6, 8, and 11 days). Values are means ± SD of five biological replicates. * Significant differences compared with the control (water-treated fruits) sample at *p* < 0.05, using Student’s test.

**Figure 9 foods-09-00892-f009:**
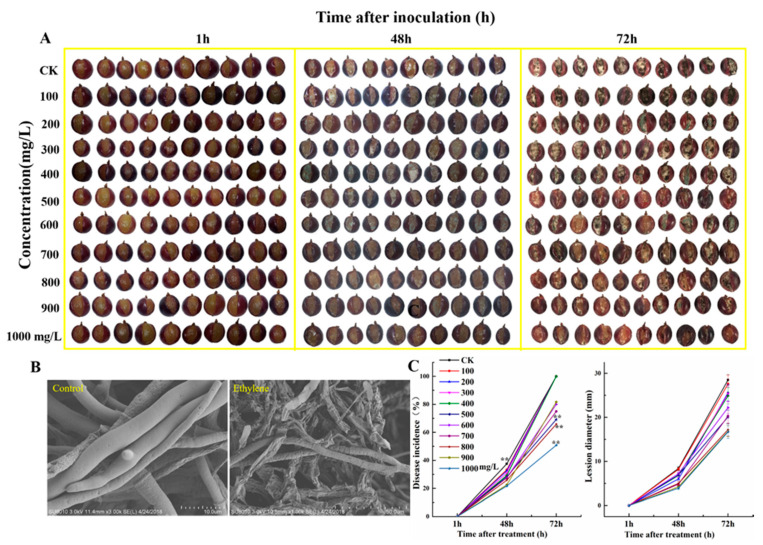
Effects of different concentrations of ethephon pretreatment on the infected grape with *B. cinerea*. (**A**) The development of *B. cinerea* on the surface of ‘Kyoho’ grape fruits in pretreated samples with different concentrations of ethephon treatment (100, 200, 300, 400, 500, 600, 700, 800, 900, 1000 mg/L) for 24 h and then inoculation with *B. cinerea*. Samples were monitored at 1, 48, and 72 h after *botrytis* infection. (**B**) The *B. cinerea* mycelium grew on the surface of the grape fruit and was observed by electron microscope at 72 h. (**C**) Fruit disease incidence and lesion diameter changes were calculated. Values are means ± SD of twenty biological replicates. ** Significant differences compared with the control (water-treated fruits) sample at *p* < 0.01, using Student’s test.

**Figure 10 foods-09-00892-f010:**
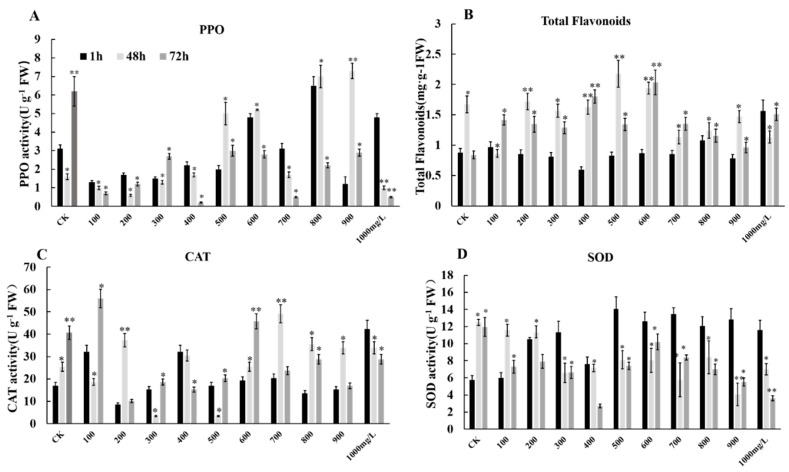
Effects of different concentrations of ethephon treatment on the total flavonoids and antioxidant enzymes. Different concentration of ethephon (100, 200, 300, 400, 500, 600, 700, 800, 900, and 1000 mg/L) were sprayed on the fruit surface for 24 h, and then *B. cinerea* was inoculated on the grape. Samples were collected at 1, 48, and 72 h after botrytis infection and measured (**A**) PPO activity, (**B**) total flavonoid content, (**C**) SOD activity and (**D**) CAT activity. Values are means ± SD of five biological replicates. * Significant differences compared with the control (water-treated fruits) sample at *p* < 0.05, ** Significant differences compared with the control (water-treated fruits) sample at *p* < 0.01 using Student’s test.

**Figure 11 foods-09-00892-f011:**
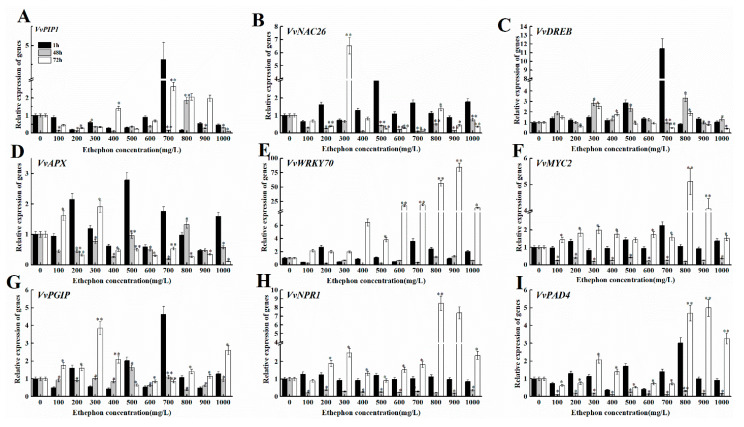
Effects of different concentrations of ethephon treatment on genes expression of disease resistance in mature grape fruit. Different concentration of ethephon (100, 200, 300, 400, 500, 600, 700, 800, 900, and 1000 mg/L) were sprayed on the fruit surface for 24 h, and then *B. cinerea* was inoculated on the grape. Samples were collected at 1, 48, and 72 h after botrytis infection and the transcript levels of disease resistant genes including (**A**) *VvPIP1*, (**B**) *VvNAC26*, (**C**) *VvDREB*, (**D**) *VvAPX*, (**E**) *VvWRKY70*, (**F**) *VvMYC2*, (**G**) *Vvpgip*, (**H**) *VvNPR1*, and (**I**) *VvPAD4*. Values are means ± SD of five biological replicates. * and ** Significant differences compared with the control (water-treated fruits) sample at *p* < 0.05 and *p* < 0.01, respectively using Student’s test.

**Figure 12 foods-09-00892-f012:**
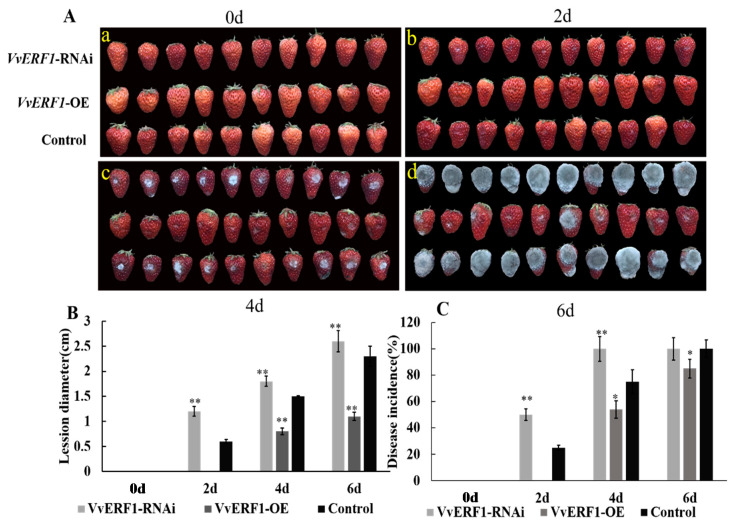
Effects of VvERF1 on strawberry fruit with transient expressions of VvERF1-OE, VvERF1-RNAi. (**A**) Strawberry fruit was inoculated with *B. cinerea* and sampled at 0, 2, 4, and 6 d. (**B**) Disease incidence. (**C**) Disease lesion diameter. Values are means ± SD of five biological replicates. * and ** Significant differences compared with the control (water-treated fruits) sample at *p* < 0.05 and *p* < 0.01, respectively using Student’s test, d: day.

**Figure 13 foods-09-00892-f013:**
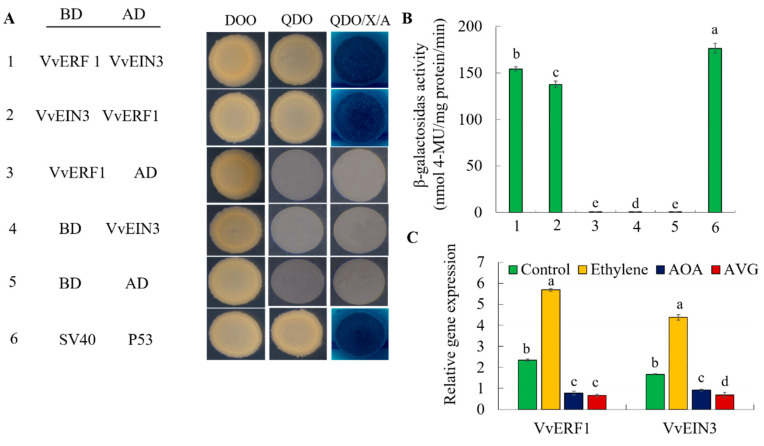
Yeast two-hybrid system for VvERF1 and VvEIN3. (**A**) VvERF1 fragments were ligated into the pGBKT7 vector (binding domain (BD)) and VvEIN3 into the pGADT7 vector (activation domain (AD)). DDO, SD medium lacking Trp/Leu; QDO, SD medium lacking Trp/Leu/His/Ade; X-a-gal, QDO medium containing x-a-gal and AbA. The SV40 and P53 genes were used as the positive control, and AD and BD vectors were as the negative control. Blue plaques indicate interaction between two proteins. (**B**) β-galactosidase activity was quantified from the sample in (**A**). (**C**) Inhibitors of ethylene biosynthesis like Aminoetoxyvinylglycine (AVG) and aminoxyacetic acid (AOA) were sprayed on the grape fruit surface respectively, and the expression levels of VvERF1 and VvEIN3 genes were measured after 3 days in grape fruits. Values are means ± SD of three biological replicates. Different letters indicated a statistical difference at *p* < 0.05 as determined by Student’s test.

**Table 1 foods-09-00892-t001:** Biomass of fungus on the VvERF1-OE and VvERF1-RNAi in strawberry fruits.

	Concentration (10^5^/mL)
Control	0.89 ± 0.19
VvERF1-OE	0.22 ± 0.14
VvERF1-RANi	1.37 ± 0.22

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
