# Peer review of "The Effect of Ethylene on the Color Change and Resistance to Botrytis cinerea Infection in ‘Kyoho’ Grape Fruits"

_foods, 2020, doi:10.3390/foods9070892_

Round 1
Reviewer 1 Report
Due to the English language use and to the lack of a robust and detailed description of the experimental plan, this paper does not reserve publication. Introduction is badly written (see after) and the experimental plan is not well detailed. On graphs reporting (*) on each point does not allow to understand to what extent averages display significant differences. However, this manuscript reports many very interesting data, apparently obtained with well-established methods. I suggest considering the possibility to write two different papers. One could refer to the ethephon effect on grapevine berry ripening; the other to the ethephon effect on Botrytis limitation, including the trial with strawberry.
English is generally very poor, particularly in paragraphs 2.1, 2.2, 2.4 and 2.5 of the M&M section.
Authors should specify if Kyoho is a winegrape or a tablegrape and discuss the role of softening in the two different circumstances.
How authors did prepare replicates? This should be explained. Did they use biological replicates in the vineyard (constituted by how many vines?) or did they collect berries (how many?) from non-identified vines in the vineyard, pooled them together and then divided into three sub-pools? Authors should describe in detail the experimental layout of the treatments.
Authors did not specify how the calculated the fragmentation index of the clusters (reported in Figure 1C). I suggest to use the same colours and symbols in all graphs (Fig. 1D has different line colours and symbols respect to 1B and 1C). In Fig. 1 legend authors state they averaged five replicates but in M&M it was reported that analysis were performed in triplicates.
Titratable acidity instead of titrate acid.
The first sentence of the abstract must be rephrased. The same for lines 20 and 21 of the abstract.
Line 25 the word ‘role’ is missing.
Introduction is a mere list of papers without a real thread among the cited papers; a true speach is missing. Moreover, references in the text are often formally incorrect.
Line 166 cinereal? Line 208 continuous ethanol?
Author Response
Dear Reviewer:
Please see the attachment

Reviewer 2 Report
Review Comments for Foods-814599
- Please look into abstract and explain. How does the strawberry suddenly appear in this sentence?
“Meantime overexpression of ethylene response factor VvERF1 improved fruit resistance against B. cinerea infection, while RNA interference of VvERF1 had an opposite effect in strawberry.”
- Abstarct is written without head and tail. It should be rewritten keeping mind to show a connection for readers interest.
- Grape (Vitis vinifera))has been classified as a non-climacteric fruit. Paper has not been discussed in light of this. By in large, non-climacteric fruit ripening is independent of ethylene, although some literature evidenced small rise of ethylene during ripening initiation in grapes.
- Following key reference from the same area are missing and must be discussed.
- Dario Cantu, Barbara Blanco-Ulate, Liya Yang, John M. Labavitch, Alan B. Bennett, Ann L.T. Powell. Ripening-Regulated Susceptibility of Tomato Fruit to Botrytis cinereaRequires NOR But Not RIN or Ethylene Published July 2009. DOI: https://doi.org/10.1104/pp.109.138701
- Polyamines Attenuate Ethylene-Mediated Defense Responses to Abrogate Resistance to Botrytis cinereain Tomato
Savithri Nambeesan, Synan AbuQamar, Kristin Laluk, Autar K. Mattoo, Michael V. Mickelbart, Mario G. Ferruzzi, Tesfaye Mengiste, Avtar K. Handa
Published February 2012. DOI: https://doi.org/10.1104/pp.111.188698
- Böttcher C, Burbidge CA, Boss PK, Davies C. Interactions between ethylene and auxin are crucial to the control of grape (Vitis vinifera L.) berry ripening. BMC Plant Biol. 2013;13:222. Published 2013 Dec 23. doi:10.1186/1471-2229-13-222
- The following title is not appropriate:
2.6. Overexpression or RNA interference and Agrobacterium-mediated infiltration in strawberry
- Discussion lack incorporation of important previous work.
Author Response

(The authors gave the same response as above.)

Reviewer 3 Report
This manuscript describes the effect of ethephon in several fruit properties as well as in the expression of different genes related to fruit quality and to Botrytis cinerea resistance/susceptibility.
The experiment desing is correct and the issue has importance to enlarge the knowledge on the fruit quality control and the postharvest tools to improve it.
However, the manuscript has several general concerns such as:
- poor description of methodology. There is missing information about repetitions used, sampling timing, etc. The methodology section needs to give every detail of the methods used in order to be able to repeat the experiment if necessary/wanted.
- Results section comments are not accurate with the results showed in the figures and tables. Some claims are very subjective with no support on statistically significant data. Each claim needs to be supported with statistical results.
- Discussion section needs to be redone. More references are needed and discussed with this manuscript´s results. Some paragraphs in this section belong to the Results section.
- English language needs extensive editing. Some sentences are difficult to understand.
Some minor comments and some referent to the general comments mentioned above can be found in the pdf document attached.

Author Response

(The authors gave the same response as above.)

Round 2
Reviewer 3 Report
The authors have carefully reviewed the corrections/suggestions made in the review process and the paper has improved significantly. Also, English has been edited. I think the manuscript is now acceptable for publication in the journal.
Author Response
Dear Reviewer: Thank you very much, it has been revised according to the comments, thank you very much for your support.